# GAIA:
# A BENCHMARK FOR GENERAL AI ASSISTANTS

**Grégoire Mialon**[1]**, Clémentine Fourrier**[2]**, Craig Swift**[3]**, Thomas Wolf** [2]**, Yann LeCun**[1]**,**
**Thomas Scialom**[1]
[1]Meta AI, [2]Hugging Face, [3]AutoGPT
{gmialon, tscialom}@meta.com, clementine@huggingface.com

## ABSTRACT

We introduce GAIA, a benchmark for General AI Assistants that, if solved, would represent a milestone in AI research. GAIA proposes real-world questions that require a set of fundamental abilities such as reasoning, multi-modality handling, web browsing, and generally tool-use proficiency. GAIA questions are conceptually simple for humans yet challenging for most advanced AIs: we show that human respondents obtain 92% vs. 15% for GPT-4 equipped with plugins. This notable performance disparity contrasts with the recent trend of LLMs outperforming humans on tasks requiring professional skills in *e.g.* law or chemistry. GAIA's philosophy departs from the current trend in AI benchmarks suggesting to target tasks that are ever more difficult for humans. We posit that the advent of Artificial General Intelligence (AGI) hinges on a system's capability to exhibit similar robustness as the average human does on such questions. Using GAIA's methodology, we devise 466 questions and their answer. We release our questions while retaining answers to 300 of them to power a leader-board hereby accessible.

## 1 INTRODUCTION

Large Language Models (LLMs) arguably open the way to general purpose systems. Indeed, the latest among them (OpenAI, 2023; Anthropic, 2023; Anil et al., 2023; Touvron et al., 2023) are fluent, knowledgeable, aligned to some extent with human preferences (Ouyang et al., 2022), and can be augmented (Mialon et al., 2023) with tools such as web browsers or code interpreters in a zero or few-shot setting (Brown et al., 2020). However, evaluating these systems is an open problem: given their emerging new capabilities, LLMs are regularly breaking AI benchmarks, at an ever-increasing rate (Kiela et al., 2023).

In search for more challenging benchmarks, current trend suggests to seek tasks that are ever more difficult for humans, and challenge LLMs with more intricate educational assessments, for example in STEM and Law, or target more complex realisations, such as writing a coherent book. But, tasks that are difficult for humans are not necessarily difficult for recent systems: the challenging MMLU or GSM8k benchmarks for example (Hendrycks et al., 2021; Cobbe et al., 2021) are already close to be solved,[1] due to rapid LLM improvement possibly combined with data contamination.[2] Furthermore, open-ended generation generally requires human or model-based evaluation (Zheng et al., 2023). Human evaluation will become less and less feasible when increasing the task complexity, *e.g.* in terms of output length or required skills: how to evaluate a book generated by an AI, or solutions to maths problems that few people in the world can solve? Model-based evaluations on the other hand are by construction dependent of stronger models hence cannot evaluate new state-of-the-art models, without mentioning potential subtle biases such as preferring the first choice presented (Zheng et al., 2023). Overall, evaluating new AI systems requires to rethink benchmarks (Chollet, 2019).

---

[1]GPT4 reportedly does 86.4% on MMLU. Human *non*-specialist accuracy on the benchmark is only 34.5% Expert-level human performance is estimated at 89.8%.

[2]See for example the case of Hellaswag.

---

### Level 1

**Question:** What was the actual enrollment count of the clinical trial on H. pylori in acne vulgaris patients from Jan-May 2018 as listed on the NIH website?
**Ground truth:** 90

### Level 2

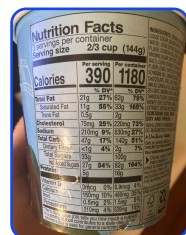

**Question:** If this whole pint is made up of ice cream, how many percent above or below the US federal standards for butterfat content is it when using the standards as reported by Wikipedia in 2020? Answer as + or - a number rounded to one decimal place.
**Ground truth:** +4.6

### Level 3

**Question:** In NASA's Astronomy Picture of the Day on 2006 January 21, two astronauts are visible, with one appearing much smaller than the other. As of August 2023, out of the astronauts in the NASA Astronaut Group that the smaller astronaut was a member of, which one spent the least time in space, and how many minutes did he spend in space, rounded to the nearest minute? Exclude any astronauts who did not spend any time in space. Give the last name of the astronaut, separated from the number of minutes by a semicolon.
**Ground truth:** White; 5876

---

Figure 1: Sample GAIA questions. Completing the tasks requires fundamental abilities such as reasoning, multi-modality handling, or tool use proficiency. Answers are unambiguous and by design unlikely to be found in plain text in training data. Some questions come with additional evidence, such as images, reflecting real use cases and allowing better control on the questions.

Alternatively to tasks that are harder for humans, AI systems could be asked to solve conceptually simple tasks yet that require accurate execution of complex sequences of actions, with large combinatorial spaces. The output could only be obtained upon successful completion of the task and be easy to validate, analogous to the Proof of Work algorithm (Jakobsson & Juels, 1999; Dwork & Naor, 1993), where a computer is asked to solve a complex problem whose solution is easy to verify. Tasks for AI assistants, given their need for access to a diverse and uncertain world, meet this criterion while being inherently rooted in practical use cases.

We move in that direction by proposing GAIA, a benchmark for General AI Assistants featuring 466 carefully crafted questions and their answer, along with the associated design methodology. Our questions are easy to create, challenging for AI systems—for LLMs, most require complex generations—, yet admit a unique, factual answer, allowing a simple and robust automatic evaluation. GAIA attempts to avoid current pitfalls of LLMs evaluation by targeting:

- Real-world and challenging questions. For example, a LLM will typically need to browse the open and changing web, handle multi-modality, or reason over multiple steps to answer our questions. Conversely, many LLM benchmarks are quite specific and/or restricted to closed and synthetic environments.

- Easy interpretability through conceptually simple tasks—non experts annotators exhibit a near perfect score—, associated reasoning trace, and few but highly curated questions. This is in contrast with aggregated benchmarks that can lack efficiency and reliability (Perlitz et al., 2023).

- Non-gameability. Answering the questions requires successful completion of some number of steps, which cannot easily be brute forced due to their diversity. The possibility to check the reasoning trace, the accuracy required in the answers, their absence in plain text from the internet prevent a possible data contamination. In contrast, multiple choice answers

(*e.g.*, MMLU) make contamination assessment more difficult since a wrong reasoning trace can more easily get to the correct choice.

- Simplicity of use. Crucially, the answers to our questions are factoid, concise and unambiguous. These properties allow simple, fast and factual evaluation. Our questions are meant to be answered in zero shot, limiting the influence of the evaluation setup. By opposition, many LLM benchmarks require evaluations that are sensitive to the experimental setup such as the number and nature of prompts (Liang et al., 2022b) (Section 8.2), or the benchmark implementation.[3]

In spite of being successful at tasks that are difficult for humans, the most capable LLMs do poorly on GAIA. Even equipped with tools, GPT4 does not exceed a 30% success rate for the easiest of our tasks, and 0% for the hardest. In the meantime, the average success rate for human respondents is 92%. Consequently, a system capable of solving GAIA can be assessed in the context of t-AGI,[4] noting that humans typically take between 6 minutes for the simplest questions to 17 minutes for the most complex ones. From a related perspective, such system would arguably be a competent General AI within the framework recently proposed in Morris et al. (2023), which also appear to be the next milestone in AI research since ChatGPT (OpenAI, 2023) is one level below. This paper covers the composition of GAIA, its design choices, and explain how to craft questions and the associated challenges so that the community can further extend the benchmark to target emerging questions such as safety associated to tool use, or multi-modality. We also analyse the successes and shortcomings of some of the most capable assistants to date, illustrating the potential of augmenting LLMs. We release a developer set of 166 annotated questions and release the remaining 300 questions without annotations: the benchmark will be notably hosted as a leaderboard. We hope our methodology will help addressing the problem of open ended generation evaluation in NLP and beyond, and believe the successful resolution of GAIA would be an important milestone towards the next generation of AI systems.

## 2 RELATED WORK

**Evaluating Large Language Models.** As LLMs capabilities have rapidly progressed, benchmarks become saturated at an increasing speed. As a example, reading comprehension was still a challenging task a few years alo (Rajpurkar et al., 2016). Wang et al. (2018) introduced the General Language Understanding Evaluation benchmark (GLUE), on which models surpassed humans within a year. Its extension (Wang et al., 2019) didn't resist for more than a couple of years after its release. More generally, with each passing year, static benchmarks are saturated and solved at human level at an ever increasing speed, as well illustrated by Kiela et al. (2023). While searching for harder evaluations, a natural direction is to explore tasks requiring professional level knowledge in various fields such as law or science: an example is MMLU (Hendrycks et al., 2021), containing over 15,000 questions covering 57 subjects across STEM, the humanities, the social sciences, and more. And yet, LLMs already passed human performance on these, and have even been reported to reach a stage where they could plausibly pass the US bar exam (OpenAI, 2023) or exceed the passing score on USMLE, a US examination program used to assess clinical competency and grant licensure (Nori et al., 2023). Directions to evaluate LLMs more holistically, on their broader conversational aspects, have included (i) compilations of evaluations (Gao et al., 2021; Liang et al., 2022a; Srivastava et al., 2023), which are often difficult to aggregate meaningfully and are prone to contamination through data leakage, (ii) human evaluation, which is time-consuming and difficult to scale, or (iii) model based evaluation to overcome this limitation (Zheng et al., 2023). However, this latter solution relies on using a more capable LLM (often GPT4) than the one currently evaluated, and the quality of the evaluation is affected by the shortcomings of the evaluator LLM, which are not always obvious and can lead to subtly incorrect results.

**Evaluating General Assistants.** While there is ongoing effort to turn Large Language Models into general-purpose assistants (see our discussion in Appendix C), appropriate evaluation is lagging

---

[3]https://huggingface.co/blog/evaluating-mmlu-leaderboard

[4]As defined in https://www.alignmentforum.org/posts/BoA3agdkAzL6HQtQP/clarifying-and-predicting-agi, a t-AGI beats, on most tasks, most human experts who are given time t to perform the task

behind. Most evaluations rely on the use of closed systems, specific API calls, and a given "correct way" to attain the answer, or simply repurpose existing evaluation datasets. ToolQA (Zhuang et al., 2023) or Gentopia (Xu et al., 2023a) for example combine existing datasets with human annotations (MMLU, MATH, etc.) at the risk of contamination during training, and without ensuring tool usage is actually tested. Gorilla (Patil et al., 2023) introduces APIBench, which tests how well an agent like system calls its specific API, similarly to API-Bank (Li et al., 2023b), which provides an API pool to help the LLM during its evaluation. AgentBench (Liu et al., 2023a) is more general, and provides a number of closed box environments inside which assistant LLMs can be deployed to answer user queries (from Unix shells to WebShopping APIs). However, because such evaluations rely on closed environments, they risk evaluating how well the assistants have learned to use specific APIs, instead of more general results grounded in real world interactions. By opposition, GAIA does not specify possible APIs, and relies on interactions with the real world. OpenAGI (Ge et al., 2023) introduces both a platform and a benchmark, made of a number of multi-steps tasks across modalities and capabilities, and is closer to our work. The core difference with GAIA is that their tasks focus on current model capabilities rather than upcoming advancements.

## 3    GAIA

This section covers the design and content of GAIA, as well as guidelines for creating questions and associated challenges.

### 3.1    A CONVENIENT YET CHALLENGING BENCHMARK FOR GENERAL AI ASSISTANTS

**What is GAIA and how does it work?**   GAIA is a benchmark for AI systems proposing general assistant questions. GAIA attempts to circumvent different pitfalls of LLMs evaluation. It is composed of 466 questions designed and annotated by humans. These questions are text-based, and sometimes come with a file (such as an image or a spreadsheet). They cover various assistant use cases such as daily personal tasks, science, or general knowledge. The questions are designed to admit a short, single correct answer, therefore easy to verify. To use GAIA, one only needs to zero-shot prompt an AI assistant with the questions and attached evidence if there are some. Scoring perfectly on GAIA requires a varied set of fundamental abilities (see Section 3.3). We provide questions along various with meta-data in supplementary material.

**Design choices.**   GAIA results both from the need for revised AI benchmarks, and the observed shortcomings of LLM evaluation.

Our first principle is to target questions that are conceptually simple although potentially tedious for humans, yet varied, rooted in the real world and challenging for current AI systems. This allows to focus on fundamental abilities such as quick adaptation via reasoning, multi-modality understanding, and potentially diverse tool use, rather than specialised skills (Chollet, 2019). The questions generally consist in finding and transforming information gathered from different and various sources, such as provided documents or the open and changing web, to produce an accurate answer. To answer the first example question above (Figure 1), LLMs should typically browse the web to find a study, then look for the correct enrolment. This goes against the trend of benchmarks that are increasingly difficult for humans, and/or operate in purely textual or artificial environments.

Our second principle is interpretability. The restricted number of highly curated questions makes the benchmark easier to use compared to aggregated ones (Perlitz et al., 2023). The conceptual simplicity of the task (human success rate is 92%) makes it easy for users to understand a model's reasoning trace. For the Level 1 question from Figure 1, the reasoning trace will mostly consist in checking the correct website, and report the correct enrolment, which is simple to verify.

Our third principle is robustness against memorization: GAIA aims to be less gameable than most current benchmarks. To complete a task, a system has to plan and successfully complete some number of steps since the resulting answer is absent by design in plain text from current pre-training data. A progress in accuracy reflects actual system progress. Due to their diversity and the size of the action space, these tasks cannot be brute-forced without cheating, for example by memorizing the ground truth. Although accidental memorization is possible through data contamination, the accuracy required in the answers, their absence from pre-training data, and the possibility to check

**System prompt:** You are a general AI assistant. I will ask you a question. Report your thoughts, and finish your answer with the following template: FINAL ANSWER: [YOUR FINAL ANSWER].

YOUR FINAL ANSWER should be a number OR as few words as possible OR a comma separated list of numbers and/or strings.

If you are asked for a number, don't use comma to write your number neither use units such as $ or percent sign unless specified otherwise.

If you are asked for a string, don't use articles, neither abbreviations (e.g. for cities), and write the digits in plain text unless specified otherwise.

If you are asked for a comma separated list, apply the above rules depending of whether the element to be put in the list is a number or a string.

**GAIA Question:** The attached Excel file contains the sales of menu items for a local fast-food chain. What were the total sales that the chain made from food (not including drinks)? Express your answer in USD with two decimal places.

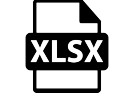

uploaded.xlsx

**GPT4 (Advanced Data Analysis mode):**

Working:

```python
import pandas as pd

# Load the Excel file
file_path = '/mnt/data/uploaded.xlsx'
df = pd.read_excel(file_path)
```

Working:

```python
# Calculate the total sales from food items (not including drinks i.e., Soda)
total_food_sales = df[
    ['Burgers', 'Hot Dogs', 'Salads', 'Fries', 'Ice Cream']
].sum().sum()

# Format the result to USD, with a dollar sign and two decimal places
total_food_sales_formatted = f"${total_food_sales:,.2f}"
```

FINAL ANSWER: $89706.00    **Ground truth:** 89706.00 ✓

Figure 2: To answer GAIA, an AI assistant such as GPT4 (here equipped with a code interpreter) needs to complete a few steps, potentially using tools or reading files.

the reasoning trace mitigate this risk. In contrast, multiple choice answers make contamination assessment difficult since a wrong reasoning trace can still get to the correct choice. If catastrophic memorization happens in spite of these mitigations, it is easy to craft new questions using the guidelines we provide in Appendix A.

Our last principle is easiness of use. Our tasks are simple prompts that may come with an additional file. Crucially, the answers to our questions are factoid, concise and unambiguous. These properties allow simple, fast and factual evaluation. Our questions are meant to be answered in zero shot, limiting the influence of the evaluation setup. By opposition, many LLM benchmarks require evaluations that are sensitive to the experimental setup such as the number and nature of prompts (Liang et al., 2022b) (Section 8.2), or the benchmark implementation.

## 3.2 EVALUATION

GAIA is designed such that evaluation is automated, fast, and factual. In practice, each question calls for an answer that is either a string (one or a few words), a number, or a comma separated list of strings or floats, unless specified otherwise. There is only one correct answer. Hence, evaluation

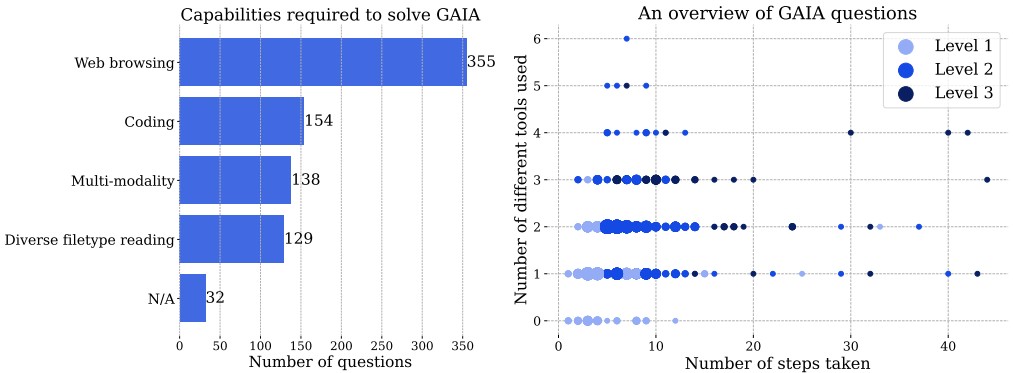

Figure 3: Left: number of questions per capability requiring at least this capability to be solved. Right: each dot corresponds to a GAIA question. At a given location, the size of the dots are proportional to the number of questions, and only the level with the highest number of questions is displayed for readability. Both figures are based on information reported by human annotators when answering the questions, and AI systems might proceed differently.

is done via quasi exact match between a model's answer and the ground truth (up to some normalization that is tied to the "type" of the ground truth). A system (or prefix) prompt is used to inform the model about the required format, see Figure 2. In practice, GPT4 level models easily follow our format. We provide our scoring function along with the leaderboard.

## 3.3 COMPOSITION OF GAIA

This subsection delves into the composition of the 466 questions we devised for GAIA. For the building process, see Appendix A.

**Capabilities coverage.** Scoring perfectly on GAIA requires advanced reasoning, multi-modality understanding, coding capabilities and generally tool use, *e.g* web browsing, for which we provide a more precise definition in Appendix E. We also include questions requiring to process varied data modalities such as PDFs, spreadsheets, but also images, videos or audio, whose distribution is reported in Appendix E (Figure 6). Figure 3 (left) is an overview of these capabilities. Although web browsing is a key component of GAIA, we do not require assistants to perform actions other than "clicks" on a website such as uploading a file, post a comment or book a meeting. Testing these capabilities in real environments while avoiding spamming websites requires careful consideration that we leave for future work, and refer the reader to recent works proposing closed environments for LLMs agents (Liu et al., 2023a). We do not provide a more detailed list of required capabilities to solve the benchmark since most questions can be solved equally well via different combinations of capabilities. For example, a given piece of evidence may have been properly memorised by an assistant LLM, or retrieved via a web search. In particular, we do not provide a fine-grained benchmarking of tool usage by LLMs, and refer the reader to Xu et al. (2023b); Li et al. (2023c).

**Increasing difficulty.** The questions can be sorted into three levels of increasing difficulty depending on the number of steps required to solve the questions, and the number of different tools needed to answer the question. There is naturally not a single definition of step or tool, and possibly many paths to answer a given question. Therefore, we rely as a proxy on the number of steps and tools used by our annotators when crafting the questions. Figure 3 (right) illustrates the distribution of our questions along these two axes. Tools are always related to one or more capability (see Appendix E). We loosely use the following definitions to attribute a level to a question:

- **Level 1** questions generally require no tools, or at most one tool but no more than 5 steps.
- **Level 2** question generally involve more steps, roughly between 5 and 10 and combining different tools is needed.
- **Level 3** are questions for a near perfect general assistant, requiring to take arbitrarily long sequences of actions, use any number of tools, and access to the world in general.

An illustration of these levels is provided in Figure 1. Those definitions are not hard constraints: for example, a question with less than 10 annotator steps but that requires complex web navigation might be categorised as Level 3 rather than 2. Our definition of the difficulty is validated in Section 4.

**Distribution of required capabilities.** While GAIA targets real-world assistant questions, we also include tasks that could potentially benefits physically impaired people, such as finding a piece of information in a small audio file. Finally, we make our best effort to cover various topic domains and cultures, although the language of the dataset is restricted to English (see Appendix B).

## 4 LLMS RESULTS ON GAIA

Evaluating LLMs with GAIA only requires the ability to prompt the model, *i.e* an API access. We use a prefix prompt before asking the model a question. To ease answer extraction, we specify a format in the prefix prompt, see Figure 2. We evaluate GPT4 (OpenAI, 2023) with and without plugins,[5] as well as AutoGPT [6] with GPT4 as backend. GPT4 currently requires to manually select plugins (see paragraph below). On the contrary, AutoGPT is able to do this selection automatically. Our non-LLM baselines are human annotators, and web search. For the latter, we type our questions in a search engine and check whether the answer can be deducted from the first page of results. This allows us to assess whether the answer to our questions can easily be found on the web or not. Whenever an API is available, we run the model three times and report the average results.

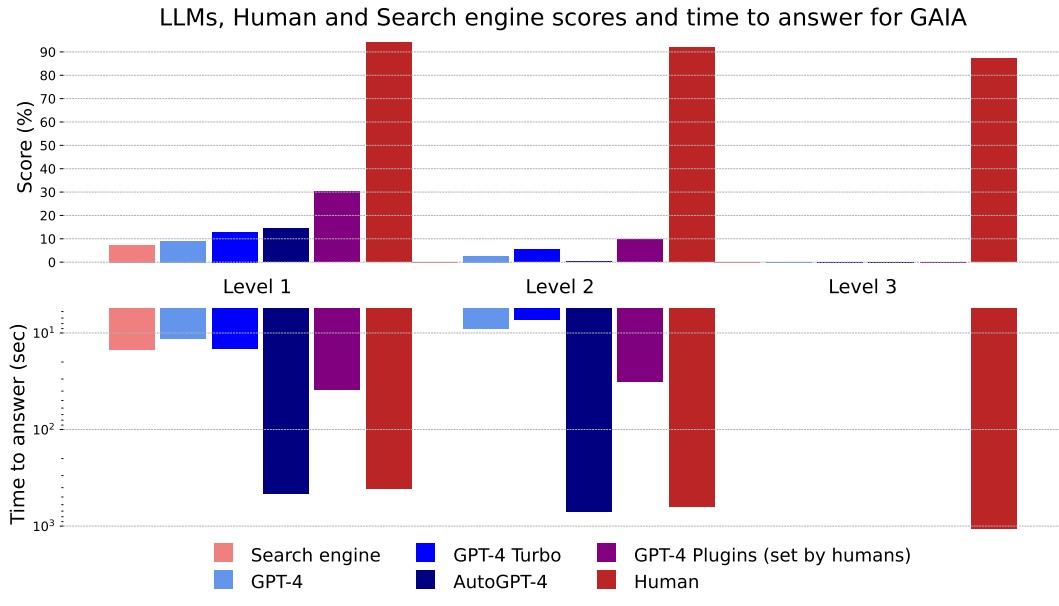

Figure 4: Scores and time to answer per method and level. As stated in the main text, GPT4 + plugins score should be seen as an oracle since the plugins were chosen manually depending on the question. Human score refers to the score obtained by our annotators when validating the questions.

**GPT4 plugins.** As opposed to GPT4, there is currently no API for GPT4 *with* plugins, and we resort to manual ChatGPT queries. At the time of the writing, the user has to manually choose between an Advanced Data Analysis mode—with code execution and file reading capabilities—, and a set of at most three third party plugins. We use either the first mode or select third parties plugins according to our best guess of the most important capabilities given the task. We often rely on (i) a tool for reading various types of links, (ii) a web browsing tool, and (iii) a tool for computation. Sadly, it is currently not possible to use a stable set of plugins over some period of

---

[5] https://openai.com/blog/chatgpt-plugins
[6] https://github.com/Significant-Gravitas/Auto-GPT, git hash of the AutoGPT version evaluated: ed172dec1947466cc0942abf75bb77b027cd433d.

time as plugins often change or disappear from the store. Similarly, the official search tool for GPT4 was removed as it could possibly circumvent paywalls, before being recently brought back. Therefore, our score for GPT4 with plugins is an "oracle" estimate of GPT4 potential with more stable and automatically selected plugins rather than an easily reproducible result.

**Results.** Our evaluation can be found in Figure 4, with more details in Table 4 (Appendix F.1). Our proposed levels of difficulty, loosely defined in terms of number of steps and number of different capabilities used, are correlated with the performance of current models, strengthening their validity. While humans excel at all levels, current best LLMs do poorly. Overall, GAIA allows to clearly rank capable assistants, while leaving a lot of room for improvement in the coming months and perhaps years.

Web search by humans might return textual results from which the correct answer can be deducted for Level 1, yet does not work when it comes to slightly more complex queries, and is also slightly slower than a typical LLM assistant since the user has to skim through the first search results. This confirms the potential of LLM assistants as competitors for search engines.

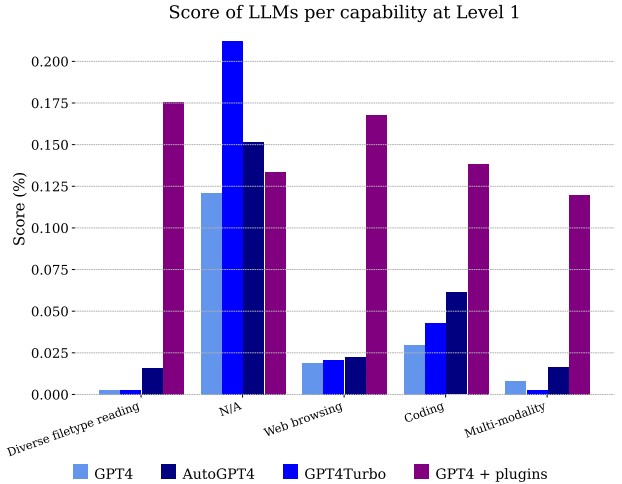

Figure 5: Score of various LLMs at Level 1 per capability. Non zero scores for non tool models for "Diverse filetype reading" and "Multi-modality" are due to tasks that can be solved differently from the way the annotators did. Non zero scores for non tool models for web browsing are mostly due to correct memorization of information required to complete intermediate steps.

The discrepancy between GPT4 results without plugins and the others demonstrate that augmenting LLMs via tool APIs or access to the web improves answer accuracy, and unlock many new use cases, confirming the huge potential of this research direction. In particular, GPT4 + plugins exhibit behaviours such as backtracking or query refinement when the result is not satisfying, and relatively long plan execution. We provide examples of such behaviours in Appendix F.1. The discrepancy with humans suggests the work needed to fully unlock this potential.

AutoGPT4, which allows GPT4 to automatically use tools, offer disappointing results for Level 2, and even Level 1 compared to GPT4 without plugins. This discrepancy might come from the way AutoGPT4 relies on the GPT4 API (prompt and generation parameters) and will require new evaluation in the near future. AutoGPT4 is also slow compared to other LLMs. Overall, the collaboration between a human and GPT4 with plugins seem to offer the best ratio of score versus time needed so far.

Figure 5 shows the scores obtained by the models splitted per capability. Unsurprisingly, GPT4 cannot deal with files and multi-modality, yet manages to solve questions for which annotators used web browsing, mostly because it properly memorised pieces of information that need to be combined to get the answer.

## 5 DISCUSSION

Designing GAIA led us to think about current and future paradigm of AI systems evaluation.

**Reproducibility for closed-source assistants.** The capabilities of models closed behind APIs might change over time (Chen et al., 2023), making an evaluation done at some point in time not reproducible. The problem can be even worse: for example, ChatGPT plugins and their capabilities change regularly, and are not accessible through ChatGPT's API yet. Reproducibility could become

even more elusive since static benchmarks might disappear in favour of benchmarks that decay through time due to their reliance on the real world. GAIA is however robust to the randomness of token generation since only the final answer, that admits a single correct response, is evaluated.

**Static versus dynamic benchmarks.** Much like other complex expert datasets, GAIA currently comes with hundreds of questions that have been carefully curated and selected. By comparison, a more massive benchmark such as MMLU has close to 15,000. Yet, MMLU consists of multiple choice questions hence is seemingly easier than our open questions. Questions that admit a single correct answer require care, and we preferred to favour quality over quantity. Moreover, we hope that our insights on question design will help the community to add more questions. GAIA is indeed likely to decay over time, be it via (i) catastrophic contamination of pre-training data or (ii) disappearance from the web of some information required to answer the questions. We are confident that the various mitigations we provide for these problems will help maintaining GAIA relevant until it is solved. Static benchmarks are broken benchmarks in the making, and making GAIA evolve year-by-year through the removal of broken questions and the addition of new ones might be an important component to better assess the generalization and robustness of AI systems.

**Towards unified evaluation of generative models.** Many GAIA tasks might be solved by calling modules that could yield errors *e.g.* an image classifier returning the wrong label. One could argue this makes evaluation ambiguous since it considers the system as a whole and does not attribute errors to sub-parts *e.g.* the web browsing or vision modules. However, the paradigm of coupling LLMs with external tools for every task beyond text understanding might not last. For example, future models might bend towards more integration between the LLM and other capabilities as in vision-language models (Alayrac et al., 2022; Laurençon et al., 2023). GAIA aims at evaluating AI systems rather than the current architectural standard. More generally, automatic, factual, and interpretable evaluation of complex generations is a long lasting problem in generative AI, another important example being images (Stein et al., 2023). Hu et al. (2023) make a step in that direction, yet rely on model-based evaluation and simple questions. Moving forward, the conjugation of multi-modal systems with GAIA might further improve advanced generative models evaluation *e.g.* image generators, via tasks requiring a complex sequence of image modifications and asking an unambiguous question on the resulting image in natural language. The answer could be found only if the modifications have been correctly applied by the model to the original image.

**Partial versus full automation.** While partial automation of a process still requires humans in the loop, full automation completely removes that need. Systems that respectively allow partial automation and full automation can be as close as a few percentage of error on a given task—the former would have say 1% and the latter 0%—, yet yield these two fundamentally different paradigms. Full automation is a goal that deep learning has been striving to achieve, without complete success to date: in spite of state-of-art results in various domains, most neural networks based systems can unpredictably fail *e.g* in common situations, impeding the advent of technologies such as self-driving cars. Solving GAIA requires full automation since no approximation is allowed in the answer. Full automation of more human activities will reshape our socio-economic landscape (Growiec, 2022), with the risk that the added value is mainly captured by the owner of the technology instead of human workers. This is a grounded argument in favour of open-source.

**On the definition of general AI assistants.** The present work focuses on creating evaluations for frontier AI systems that are more challenging than current popular benchmarks, and take into account the latest capabilities of these models. While we consider any system general enough to solve GAIA to be *de facto* an excellent AI assistant, more work is required to agree with the community on the skills and level of expertise that would more rigorously define a truly general AI assistant.

## ACKNOWLEDGEMENTS

The authors would like to thank Nicolas Usunier for suggesting the web search baseline, Edwin Chen for helping us improve our unusual protocol for annotators, Yacine Jernite for sharing his insights on diversity when benchmark building, and Sasha Luccioni for taking the time to proofread some sections where proper English was eluding us.

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

## A  BUILDING AND EXTENDING GAIA

This subsection delves into our question design and annotation process. In particular, we discuss some associated challenges and hope our insights will help the community building overGAIA.

**Crafting questions.**    Our questions are created by humans[7] and aim to reflect realistic use cases of AI assistants. The authors designed initial questions, and gave them as examples to annotators along with instructions (reported in Appendix F) to create more questions. The questions were based on one or more sources of truth that were often specified in the question to avoid ambiguity. Examples of sources of truth are trusted web pages that have low chance to disappear anytime soon *e.g.*, Wikipedia, Papers With Code, or arXiv. In other cases, the source of truth is entirely provided with the question, *e.g.*, an attached document. The last case is a self-contained question, *e.g.*, a small puzzle. We do not specify a fixed list of sources of truth in order to enforce question diversity and avoid memorisation. Apart from puzzles, most questions were created by finding and potentially combining information from different sources of truth to produce a specific answer. Once a question was created, it was also annotated, *i.e.* the question creator provided an answer as well as meta-data: which tools were needed, which steps were taken, or how many time was required to answer. A typical annotation result is presented in Table 1 (Appendix E).

**Validating questions.**    Most of the work associated with crafting questions consists in ensuring that they are unambiguous, *i.e.*, there is a single correct answer. This property allows fast and factual evaluation, hence it is crucial to maintain it. Ambiguities can be subtle and rarely obvious to the creator of a question. For example, a question is ambiguous if it does not specify a version for a web page while the information needed to answer the question is different in other versions. We therefore asked two new annotators to independently answer each question. If the original annotator and the two new annotators arrived at the same answer, the question was validated. Questions on which annotators disagreed generally only required a simple fix, but were removed otherwise. For this reason, question creation can hardly be automated while keeping the interest and variety of questions high. We report statistics on this validation phase in Table 3 (Appendix E). 68% of the questions were good as is, while the rest had to be corrected or removed. While the questions are conceptually simple, annotators might do inadvertent mistakes: we estimate the annotator's success rate to be 92% when aggregated on all levels of difficulty, and report this as the human score for GAIA. It is close to perfect, demonstrating that GAIA is simple for non experts. We estimate the creation of a question, including its validation by two supplementary annotators and potential repairs, to require two hours of annotator time.

**Challenges associated to relying on the web.**    Designing questions can be delicate when a source of truth is hosted on the web. First, the evidence might change over time. For example, a Wikipedia article could be updated between the moment the question is created and the moment it is asked to an AI assistant, potentially removing the evidence required to answer. For such questions, it is often important to specify a version of the evidence, such as the page's date. In practice, we find our benchmark to be robust to these changes since we try to rely as much as possible on evidence that will likely pass the test of time. Second, some website owners wish to prevent access to parts or totality of their website from bots via their `robots.txt` files. While this is rather a demand than a constraint, it is obviously desirable to comply. For example, OpenAI provides instruction to website owners wishing to forbid access to GPT4 on how to modify their `robots.txt` accordingly. Hence, we verify that accessing the part of the website hosting the evidence is not restricted.

## B  LIMITATIONS

While GAIA attempts to circumvent current pitfalls of LLM benchmarks, some limitation remains.

**Missing evaluations.**    In its current form, GAIA does not evaluate the trace leading to the answer. Indeed, as opposed to the ground truth which is unique, different paths could lead to the correct answer and there is no obvious and simple ways to grade those, while we prioritized easiness of use for GAIA. Going forward, human and model-based evaluations, albeit limited, are interesting

---

[7]More precisely, in a collaboration between our teams and compensated annotators from Surge AI.

options to evaluate the plans, and could be quite convenient since (i) our questions rarely require expert knowledge, thus alleviating the need to find specialized annotators, and (ii) the judge can rely on the ground truth: it is often faster to verify than to independently derive the answer. We leave the addition of human and model-based evaluation for future work. Finally, we only evaluate the strongest available LLMs that have access to tools hence are able to obtain informative scores. However, OpenAI's API does not provide the detailed log of tool calls yet, which would be required for fine-grained analysis. We look forward to add other models with sufficient tool using capabilities and logging, especially in open source.

**On the cost of designing unambiguous questions.** The price to pay for a real-world yet easy to use benchmark corresponds to making sure the questions are unambiguous. We find that two rounds of annotations are required, a first annotator making their best effort to design an unambiguous question—wich takes more time than *e.g.* ranking two different generations for RLHF—, and two supplementary annotators independently answering the question and disambiguating it if necessary. In spite of this thorough process, possible ambiguities remain. However, the annotation cost is fixed and probably small compared to the potential cost of multiple untrustworthy evaluations. A question might be ambiguous for a perfectly logical computer yet not ambiguous for humans: this is not a problem since we want AI systems to be aligned with human preferences. We believe human annotators are currently essential to have diverse and grounded questions, as opposed to programmatically generated ones. A similar argument is made in Chollet (2019). One could however synthetically generate GAIA-like data by relaxing the unambiguity constraint, *e.g.* for training purpose. Additionally, some GAIA questions come with many details hence seem unnatural: these details ensure the question admits only one correct answer and are therefore necessary. In practice, a user would ask an under-specified question, and a useful assistant would answer by citing its sources or keeping the most trustworthy one. Both are difficult to factually evaluate, and we leave that aspect for future work.

**Lack of linguistic and cultural diversity.** A big limitation of GAIA is its lack of language diversity: all questions are asked in "standard" English only, and many questions mostly rely on English web pages. This benchmark will therefore not validate the usefulness of assistants for non-English speakers (80% of the global world population), their usefulness on the non English-speaking web (about half of its content), nor on any sort of dialectal variation of English. As such, GAIA is only a first step to estimate the potential of AI assistants, but should not be seen as an absolute general proof of their success. We hope to fill this gap in future work or through community involvement.

## C  EXTENDED RELATED WORK.

**Large Language Models as General Assistants.** Several avenues have been explored to turn LLMs into general-purpose assistants: (i) using single agent LLMs with better capabilities through Chain of Thought prompting or equivalent mechanisms, such as GPT-Engineer (Osika, 2023), AutoGPT (Yang et al., 2023); (ii) using multiple agent LLMs to debate and together reach better conclusions to answer user queries (Li et al., 2023a; Hong et al., 2023; Chan et al., 2023; Talebirad & Nadiri, 2023); (iii) using single agent LLMs augmented with specific tools, such as Blender Bot 3 (Shuster et al., 2022), BOLAA (Liu et al., 2023b) and AssistGPT (Gao et al., 2023) extending LLMs with planning components, Socratic Models (Zeng et al., 2022) or Visual ChatGPT (Wu et al., 2023) extended with multimodal models, WebGPT Nakano et al. (2021) fine-tuned for web-search, or a collection of tools and APIs, such as Toolformer (Schick et al., 2023) fine-tuned for general tool usage, ViperGPT (Surís et al., 2023) using coding capabilites to generate correct API calls, HuggingGPT (Shen et al., 2023) leveraging calls to the HuggingFace ecosystem to extend its LLM with other ML models capabilities, or even (iv) providing full new API/tooling libraries, such as the OpenAI plugins, SemanticKernel (Microsoft, 2023), Langchain (Chase, 2022) and MiniChain (Rush, 2023).

## D  DATACARD

We follow (Bender & Friedman, 2018) for the creation of this datacard, where we try to summarise and centralise all information which might be relevant for analysis of this dataset.

**Curation rationale.** This is detailed in Appendix A and Appendix F.

**Language variety.** Information about our annotators' nationality was not provided, but they were all based in the US, and all questions, answers, and meta-data were written in mainstream English (therefore most likely en-US). We can also note that all authors of this paper are French and do not have English as a first language, which might have lead to the inclusion of non-standard English phrasing in the questions or answers.

**Curators and Annotators demographic.** Following the definitions proposed in (Bender & Friedman, 2018), building GAIA required the work of Curators, who devised the questions and their answer, and Annotators, who independently annotated the questions to assess their non-ambiguity. Both come from the following population:

- **Age**:
  - 18-25: 17%
  - 26-35: 39%
  - 36-45: 26%
  - 45-55: 13%
  - 56-65: 4%
- **Gender**: 57% Male, 43% Female.
- **Academic background**:
  - Bachelor's Degree: 61%
  - Master's Degree: 26%
  - PhD: 17%

**Text characteristics.** This is detailed in Appendix E.

## E EXTENDED DESCRIPTION OF GAIA.

**Description of capabilities.** When answering the questions, annotators specified the steps that were followed and listed the tools they use. Based on the set of tools that were mentionned by the annotators, we defined capabilities required by GAIA. For each capability, we report examples of corresponding tool as reported by annotators.

- **Web browsing**: tools related to search the web and browse websites. Examples: `Web browser, Search engine, Website widget access, Access to YouTube, Google Street View`.
- **Multi-modality**: tools related to understanding data modality other than text. Examples: `A speech-to-text tool, Video recognition, Image recognition, OCR, Google Street View`.
- **Coding**: tools related to code execution. Examples: `Python, a calculator, Substitution cipher encoder, C++ compiler, A word reversal tool / script`.
- **Diverse filetype reading**: tools related to understanding various type of files given by a user or found on the web. Examples: `PDF viewer, Excel file access, PowerPoint viewer, CSV access, Txt file access`.
- **N/A**: tools for tasks that can currently be performed by non-augmented LLMs. Examples: `Tetris rules database, German translator, Spell checker, Text Editor, Bass note data`.

Note that a tool can belong to different categories. For example, `Google Street View` requires access to the web, browsing, but also multi-modality. Hence, these categories are indications of the capabilities required by GAIA and not a perfect typology of our questions.

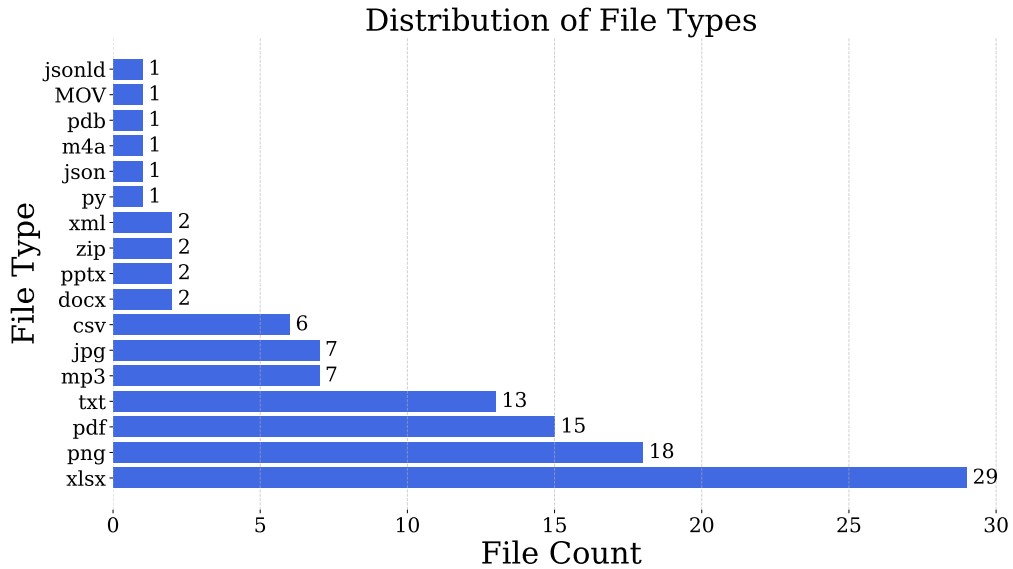

Figure 6: Initial distributions of file types in GAIA.

**Filetypes.** Some GAIA questions come with additional files, whose distribution is given in Figure 6.

**Difficulty of the questions.** Our analysis of the time taken by the annotators to answer a question shows a correlation with the number of steps taken. The correlation is less clear with the number of different tools used to answer.

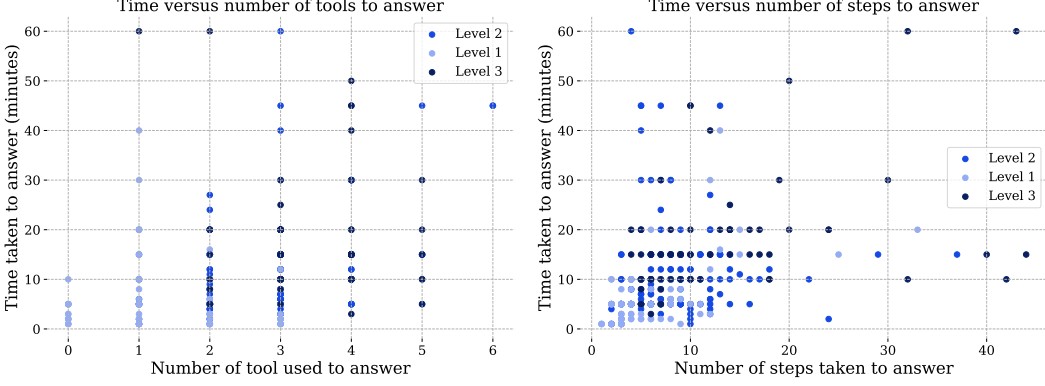

Figure 7: Using multiple tools does not necessarily involve more time to answer a question.

Figure 8: Unsurprisingly, the number of steps taken to answer is correlated to the time taken.

## F    EXTENDED DESCRIPTION OF OUR QUESTION DESIGN FRAMEWORK.

**Question creation phase.** We provided the annotators with a seed set of GAIA questions we devised ourselves, accompanied with the following instructions:

We want to augment the dataset of provided questions (not variations of what we already have).

Requirements:

- Make sure your question is based on a source of truth (Wikipedia, arXiv, githhub, other...). For Level 2 and Level 3, a good way to create questions is to combine sources of truth.

| Question | What was the actual enrollment count of the clinical trial on H. pylori in acne vulgaris patients from Jan-May 2018 as listed on the NIH website? |
|---|---|
| File | None |
| Level | 1 |
| Steps | - Searched "nih" on Google search.
- Clicked the top link to nih.gov.
- Searched "h pylori acne" in the search box.
- Clicked "More" and selected "Clinical Trials".
- Clicked the result about H. Pylori and acne.
- Checked the date to confirm it was January to May 2018.
- Opened "Tabular View".
- Scrolled down to Actual Enrollment and recorded the number. |
| Number of steps | 8 |
| Answer | 90 |
| Time to answer | 8 minutes |
| Tools | - Web browser |
| Number of tools | 1 |

Table 1: An annotated question during the question creation phase.

- Make sure the answer to your question does not exist on the internet in plain text.
- Make sure the answer to your question is a number or at most a few words to make evaluation robust.
- Make sure the answer to your question does not change with time. This includes potential deletion of the source of truth.
- Make sure the answer to your question is unambiguous.
- Make sure your question is "interesting", *i.e.* by reading it you think that an AI assistant answering this kind of question would help you a lot.
- Make sure your question can be answered in a reasonable amount of time by a human annotator.
- *(Added later on)*: check the `robots.txt` of the website containing the information needed to answer so that it is accessible to AI assistants.

The annotators were also asked to answer the questions they created. We provide a typical example of annotated question in Table 1.

**Validation phase.**    After question creation, we ask two new independent annotators to answer the questions to check it is not ambiguous. We provide a typical annotator output for the validation phase in Table 2, as well as additional statistics on the validation phase of our protocol in Table 3. If the new annotators don't fully agree with the original answer and there is no human error, the question is repaired if possible and removed otherwise.

We estimate the creation of a question, including its validation by two supplementary annotators and potential repairs, requires two hours of annotator time.

### F.1   EXTENDED EVALUATION

We provide the detailed scores of the different methods evaluated in Table 4.

| Question | What was the actual enrollment count of the clinical trial on H. pylori in acne vulgaris patients from Jan-May 2018 as listed on the NIH website? |
|---|---|
| File | None |
| Level | 1 |
| Verifier response | 90 |
| Answer match | Yes - my answer matches the correct answer. |
| Cause of mismatch | None |

Table 2: An annotated question during the validation phase.

| After two new, independent annotators answer for all crafted questions: | |
|---|---|
| Two new annotators agree with original answer | 55% |
| One new annotator agree with original answer, other disagree | 27% |
| Two new annotators disagree with original answer | 18% |
| Valid questions (aggregated)* | 68% |
| Valid Level 1 questions | 75% |
| Valid Level 2 questions | 68% |
| Valid Level 3 questions | 47% |
| Human score (aggregated)** | 92% |
| Human score for Level 1 | 94% |
| Human score for Level 2 | 92% |
| Human score for Level 3 | 87% |

Table 3: Statistics on the validation phase. 623 newly crafted questions were validated by two new annotators each. The statistics were computed on their 1246 annotations. *: a valid question is a question for which two annotators give the same answer as the question designer, or only one annotator gives the same answer as the question designer and the other made a mistake. **: the human baseline is computed as the fraction of correct answers for all tentative on valid questions by the new annotators.

| Metric | Score in % (↑) | | | Avg. time to answer in mins (↓) | | |
|---|---|---|---|---|---|---|
| Level | Level 1 | Level 2 | Level 3 | Level 1 | Level 2 | Level 3 |
| Number of questions | 146 | 245 | 75 | 146 | 245 | 75 |
| GPT4 | $9.1 \pm 2.5$ | $2.6 \pm 0.6$ | 0 | 0.19 | 0.15 | N.A. |
| GPT4 Turbo | $13.0 \pm 2.1$ | $5.5 \pm 1.4$ | 0 | 0.24 | 0.12 | N.A. |
| AutoGPT (GPT4 backend) | 14.4 | 0.4 | 0 | 7.6 | 11.7 | N.A. |
| GPT4 + plugins* | 30.3 | 9.7 | 0 | 0.65 | 0.53 | N.A. |
| Search engine | 7.4 | 0 | 0 | 7.4 | N.A. | N.A. |
| Human annotator** | 93.9 | 91.8 | 87.3 | 6.8 | 10.5 | 17.7 |

Table 4: Score and average time to answer for various baselines on GAIA in %. *: GPT4 + plugins scores were obtained by manually selecting plugins, and cannot be reproduced exactly for the reasons described in the main text. **: Human score corresponds to the portion of correct answers by validation annotators for valid questions. Whenever we have direct API access, we run the model three times and report the average. Times for APIs were obtained by running the API on 20 questions then averaging, and were taken at a single point in time: they aren't meant to reflect GPT4 vs. GPT4 Turbo speed, but GPT4 vs. other types of methods for answering GAIA.

**Additional examples of GPT4 answering GAIA.** We provide more reasoning traces of GPT4 with and without plugins when answering GAIA. The output of AutoGPT is currently much longer, denser and less interpretable thant GPT4. Examples of AutoGPT outputs are therefore provided in the supplementary material for the same GAIA question as the example in Figure 9.

**System prompt:** You are a general AI assistant. I will ask you a question. Report your thoughts, and finish your answer with the following template: FINAL ANSWER: [YOUR FINAL ANSWER]. YOUR FINAL ANSWER should be a number OR as few words as possible OR a comma separated list of numbers and/or strings. If you are asked for a number, don't use comma to write your number neither use units such as $ or percent sign unless specified otherwise. If you are asked for a string, don't use articles, neither abbreviations (e.g. for cities), and write the digits in plain text unless specified otherwise. If you are asked for a comma separated list, apply the above rules depending of whether the element to be put in the list is a number or a string.

**GAIA Question:** Your final answer should be a number, or as few words as possible. Where were the Vietnamese specimens described by Kuznetzov in Nedoshivina's 2010 paper eventually deposited? Just give me the city name without abbreviations.

**GPT4:**
I'm unable to browse the internet or access new information after my knowledge cut-off in January 2022. Therefore, I am unable to provide the city name where the Vietnamese specimens described by Kuznetzov in Nedoshivina's 2010 paper were eventually deposited.

FINAL ANSWER: `Unable to provide`   **Ground truth:** `Saint Petersburg` ✗

---

**System prompt:** You are a general AI assistant. I will ask you a question. Report your thoughts, and finish your answer with the following template: FINAL ANSWER: [YOUR FINAL ANSWER]. YOUR FINAL ANSWER should be a number OR as few words as possible OR a comma separated list of numbers and/or strings. If you are asked for a number, don't use comma to write your number neither use units such as $ or percent sign unless specified otherwise. If you are asked for a string, don't use articles, neither abbreviations (e.g. for cities), and write the digits in plain text unless specified otherwise. If you are asked for a comma separated list, apply the above rules depending of whether the element to be put in the list is a number or a string.

**GAIA Question:** Your final answer should be a number, or as few words as possible. Where were the Vietnamese specimens described by Kuznetzov in Nedoshivina's 2010 paper eventually deposited? Just give me the city name without abbreviations.

**GPT4 (Web browsing):**
Browsing Bing:

> 🔍 "Vietnamese specimens described by Kuznetzov Nedoshivina 2010 paper deposition location"
>
> ☰ Reading content

Result:

> A catalogue of type specimens of the Tortricidae described by V. I. Kuznetzov from Vietnam and deposited in the Zoological Institute, St. Petersburg : Nedoshivina, Svetlana : Free Download, Borrow, and Streaming : Internet Archive

FINAL ANSWER: `Saint Petersburg`   **Ground truth:** `Saint Petersburg` ✓

Figure 9: Proper web search is very effective to answer GAIA questions. The reasoning trace was obtained with the current GPT4 web browsing version.

**System prompt:** You are a general AI assistant. I will ask you a question. Report your thoughts, and finish your answer with the following template: FINAL ANSWER: [YOUR FINAL ANSWER].
YOUR FINAL ANSWER should be a number OR as few words as possible OR a comma separated list of numbers and/or strings.
If you are asked for a number, don't use comma to write your number neither use units such as $ or percent sign unless specified otherwise.
If you are asked for a string, don't use articles, neither abbreviations (e.g. for cities), and write the digits in plain text unless specified otherwise.
If you are asked for a comma separated list, apply the above rules depending of whether the element to be put in the list is a number or a string.

**GAIA Question:** In the film Goldfinger, what color was the object that James Bond concealed himself and his companion Pussy Galore at the end of the film? If there are multiple colors, put them in a comma-separated list in alphabetical order.

**GPT4 (Web browsing):**

Browsing Bing:

> 🔍 "Goldfinger ending scene object color"
>
> 🔖 Goldfinger (1964) ending / spoiler
>
> ☰ Reading content
>
> 🔍 "Goldfinger ending scene parachute color"
>
> ☰ Reading content

FINAL ANSWER: `Orange, White`    **Ground truth:** `orange, white` ✓

Figure 10: The official GPT4 browsing plugin is able to refine its initial query based on the information it finds. This reasoning trace was obtained with the GPT4 browsing plugin version that was removed, and could not be reproduced with the new version.

**System prompt:** [Same system prompt as Figure 10.]

**GAIA Question:** A standard Rubik's cube has been broken into cubes making up its sides. The cubes are jumbled, and one is removed. There are 6 cubes with one colored face, 12 edge cubes with two colored faces, and 8 corner cubes with three colored faces. All blue cubes have been found. All cubes directly left, right, above, and below the orange center cube have been found, along with the center cube. The green corners have all been found, along with all green that borders yellow. For all orange cubes found, the opposite face's cubes have been found. The removed cube has two colors on its faces. What are they? Answer using a comma separated list, with the colors ordered alphabetically.

**GPT4:**
To solve this, let's break it down step by step and visualize a standard Rubik's cube:

- A standard Rubik's Cube has 6 faces: Front, Back, Left, Right, Up, Down.
- Each face has a center cube (which determines the color of that face), surrounded by 4 edges and 4 corners.
- Each edge has two colors, and each corner has three colors.

Cube Structure:

- Orange: Front
- Red: Back
- Blue: Left
- Green: Right
- White: Up
- Yellow: Down

Step-by-Step Analysis:

- All blue cubes have been found.
- All cubes directly left, right, above, and below the orange center cube have been found, along with the center cube.
- This means the central, top, bottom, left, and right orange cubes have been found.
- All green corners have been found, along with all green that borders yellow.
- This means all green-yellow edges and corners have been found.
- For all orange cubes found, the opposite face's cubes have been found.
- This means all the red cubes opposite the found orange cubes have been found.

Solution Approach: Since the removed cube has two colors on its faces, it must be an edge cube. To determine which one is missing, we should look for the edge cube that is not accounted for by the given conditions.

- All blue cubes found, which means all blue edges and corners are found.
- The orange center, and all surrounding it, are found, and hence the opposite reds are found too.
- All green corners are found, and green that borders yellow are found too.
- By deducting the cubes found from the total cubes, we will find the missing cube, which is the edge cube between the red and yellow faces.

FINAL ANSWER: `Red, Yellow`     **Ground truth:** `green, white` ✗

Figure 11: GPT4 and other assistants struggle on puzzles, which often are Level 1 questions.

