# OpenReview forum: "GAIA: a benchmark for General AI Assistants"
_ICLR.cc/2024/Conference — ICLR 2024 poster_

### Official Review · Reviewer_GSBU · 2023-10-31

**Soundness:** 3 good
**Presentation:** 4 excellent
**Contribution:** 3 good
**Rating:** 8
**Confidence:** 4

**Summary:**

The paper introduces a novel dataset, GAIA, intended for evaluating general AI assistants. The tasks within the dataset are designed to be easy for humans, but challenging for state-of-the-art models. These tasks necessitate a variety of skills including web browsing, tool use, and reasoning. Furthermore, the answers to these problems are simple and easy to evaluate.

**Strengths:**

1. The design principles of GAIA are commendable and innovative, providing a comprehensive framework for creating similar datasets. This is the type of dataset that would ideally test all AI assistant models.
2. The procedures for creating the dataset are reliable and well-structured, utilizing human input to generate questions and validate the quality of the dataset.
3. The paper is well-written, providing clear and understandable content.

**Weaknesses:**

1. The dataset's size is relatively small, which might limit its applicability.
2. The paper does not provide sufficient assurance regarding the coverage or diversity of the problems within the dataset.

**Questions:**

1. What are the main challenges that GAIA presents to current models/systems? Do these challenges stem from incorrect tool usage or mistakes made during step-by-step execution? Can you provide an error analysis of the current models?
2. The paper mentions that "level 3 are questions for a near-perfect general assistant." How do you define a perfect general assistant, and how do you ensure that good performance on your dataset (at level 3) implies a perfect general assistant? These questions could help determine if there are other abilities not covered by your dataset.
3. What is the distribution of questions across the three levels?
4. It would be helpful to have a detailed table showing the performance of each model, including the total score for all three levels.

---

> ### Author Response · Authors · 2023-11-20
>
> We thank the reviewer for their insightful comments.
>
>
> ## Detailed error analysis
>
> We discuss this point, which is shared with other reviewers, in our general answer.
>
>
> ## On perfect general assistants
>
> We think about Level 3 as questions requiring arbitrarily many tools and steps (albeit staying conceptually simple for humans). We dub an assistant solving this level as “perfect” since there is theoretically no upper bound for the number of tools and steps. In addition, while the number of steps and tools required grows linearly between levels, the associated difficulty grows exponentially since one more step means many new possible mistakes.
>
>
> ## Distribution of questions and detailed performance table
>
> There are 146 Level 1 questions, 245 Level 2 questions, and 75 Level 3 questions. The aggregated scores of each model is therefore as follows:
>
> |               | Level 1 | Level 2 | Level 3 | Aggregated |
> |---------------|---------|---------|---------|------------|
> |Number of questions|146|245|75|466|
> |Model and scores |||||
> | Search Engine |    7.4     |     0    |     0    |     2.3       |
> | GPT4          |     6.4    |     2.5    |    0     |   3.3         |
> | AutoGPT4      |   14.4      |  6.5       |     0    |    7.9        |
> | GPT4 + plugins|  30.3       |    9.7     |     0    |     14.6       |
> | Human |  93.9       |    91.8     |     87.3    |   91.7        |
>
> We will update Table 4 in the revised manuscript to include this information.

---

> > ### Comment · Reviewer_GSBU · 2023-11-21
> >
> > Thank you for your response. I would maintain the original score of 8: accepted, as it is a quality paper.

---

### Official Review · Reviewer_FuKx · 2023-11-01

**Soundness:** 3 good
**Presentation:** 3 good
**Contribution:** 3 good
**Rating:** 8
**Confidence:** 3

**Summary:**

This paper proposes Gaia, a general purpose benchmark for image+text assistants. The questions generally require web searching and thus doing retrieval over a lot of documents with images. A few things:

* There are 466 highly curated questions in the dataset
* the questions are associated with 'levels' corresponding to the # of tools/steps required
* at creation time, the answers can be found by looking at websites that don't have bots banned via robots.txt files
* the creation process is crowdsourced and meant to focus on tool use


The paper evaluates GPT4, autogpt4, GPT4+plugins and includes human performance. Note that this is GPT4 without vision so it can't do any of the image questions, nor can it do retrieval on documents with images.

----

Update: I read the other reviews and the rebuttal, thanks (and thanks for promising to add GPT + vision results!). I still support this paper and recommend it be accepted; I can't find a compelling reason from the negative review (I think ICLR seems like a fine venue for this paper).

**Strengths:**

To this reviewer, creating harder benchmarks that aren't (as) gameable is an important research direction. The ideas proposed seem interesting and probably important for the community to discuss further. E.g. what _is_ actually important for a multimodal assistant to have? which tools should it be able to use? etc etc. The idea of making it hard by requiring open-ended internet use seems like a novel contribution to this reviewer. The open-endedness should hopefully make it harder versus older multihop datasets like MS-Marco, etc. that already provide a list of optional passages to reason over.

**Weaknesses:**

I think the paper could use more details around how the benchmark works and was constructed, but that's stylistic preference. as a result I have some questions, will update my score if they can be resolved --

**Questions:**

My main concern is about evaluation (wasn't sure whether to put these under 'questions' or 'weaknesses' as there's significant ambiguity here on my end:

* How are humans scored? The paper writes "Human score corresponds to the portion of correct answers by validation annotators for valid questions." Are these the same validators involved for validating the data or a new fresh set of annotators? This is important to this reviewer as there are many datasets that claim suspiciously high human performance because they didn't run validation with a new set of annotators.
* How do you grade open-ended answers? e.g. "St. Petersburg" vs "Saint Petersburg". I couldn't tell from the paper how the correctness is actually determined.

I'd also be curious as to the portion of answers that GPT4 can't solve right now because it doesn't have access to the right tools (e.g. image understanding)?, versus using those tools well. Similarly I'd be curious if GPT4 with vision can answer those questions.

---

> ### Author Response · Authors · 2023-11-20
>
> We thank the reviewer for their insightful comments.
>
>
> ## Details on human score
>
> The set of annotators used to measure human performance is different from the set that created the questions and there is no second try, hence we believe this score is reliable. To clarify, our process is as follows:
> - First, an annotator A creates a question and reports its answer.
> - Then, the question is given to 2 new annotators B and C to answer. B does not see A and C’s answer, and similarly for C. There are two cases:
>     - If the answers from annotators A, B, and C are similar the question is correct and the humans scored it well. We count B and C as correct answers for this question.
>     - If the answers are different, the annotators assess whether this was due to an ambiguity in the question or a human error, and there are again two cases:
>          - The question was ambiguous, e.g., “How many layers does BERT have?” and one Reviewer answered 6 referring to BERT-small while the others said 12 referring to BERT-base. We do not use it to compute a human score.
>          - The question was clear and an annotator, B or C, made a clear mistake when answering it. We count it as a human error.
>
>
> We hope this clarification solves the reviewer's legit concerns.
>
>
> ## Details on grading open ended answers
>
> Thanks for raising this important point, we will add the following paragraph on evaluation in the manuscript:
>
> “In practice, each question is formulated so that the correct answer is unique and is formatted to be either a string (one or a few words), a number, or a comma separated list of strings or floats, unless specified otherwise. There is therefore only one correct answer. Hence, evaluation can be done via exact match between a model's answer and the ground truth (up to some simple normalization that is tied to the “type” of the ground truth). A system (or prefix) prompt is used to inform the model about the required format. In practice, GPT4 level models easily follow our format.”
>
> Prefix prompt: *You are a general AI assistant. I will ask you a question. Report your thoughts, and finish your answer with the following template: FINAL ANSWER: [YOUR FINAL ANSWER].
> YOUR FINAL ANSWER should be a number OR as few words as possible OR a comma separated list of numbers and/or strings.
> If you are asked for a number, don't use comma to write your number neither use units such as $ or percent sign unles specified otherwise.
> If you are asked for a string, don't use articles, neither abbreviations (e.g. for cities), and write the digits in plain text unless specified otherwise.
> If you are asked for a comma separated list, apply the above rules depending of whether the element to be put in the list is a number or a string.*
>
> When needed, the question also details the format. Importantly, some examples in our submission including the one mentioned by the reviewer mistakenly come from an older version of the dataset. In the current GAIA version, the prefix prompt and, when needed, the question asks to not use abbreviations. Hence, the question in the manuscript is actually:
>
> *Where were the Vietnamese specimens described by Kuznetzov in Nedoshivina's 2010 paper eventually deposited? Just give me the city name without abbreviations.*
>
> St. Petersburg would actually be incorrect in the current version. We fixed this in the manuscript.
>
> Finally, we provide our scoring function in the supplementary material.
>
>
> ## Detailed error analysis for GPT-4
>
> We discuss this concern, that is shared with another reviewer, in the general comment.
>
>
> ## Results for GPT4-vision
>
> GPT4-vision was not available at the time of the writing, and the API is currently limited to 100 requests per day and organization. We will add GPT4-vision results by the end of the rebuttal if possible (it will require at least a few days to run because of OpenAI’s limitation), and commit to add it in the final version of the paper.

---

> > ### Comment · Reviewer_FuKx · 2023-11-23
> > **thanks!**
> >
> > thanks! I think these clarifications + the GPT4-vision results will improve the paper. I still vote it be accepted.

---

### Official Review · Reviewer_CDvr · 2023-11-01

**Soundness:** 3 good
**Presentation:** 4 excellent
**Contribution:** 3 good
**Rating:** 8
**Confidence:** 5

**Summary:**

The authors propose a new benchmark for General AI assistants, called GAIA. The tasks require complex reasoning and tool usage skills (e.g. web browsing, integrating information from different modalities, and so on). The data is human-generated and human-validated. The resulting benchmark is highly challenging for current LLMs while humans perform quite well on it.

**Strengths:**

- The paper clearly identifies its goals and places itself in the context of previous research. I especially appreciated how the authors broke away from the common thread of evaluating AIs on tasks that are more and more challenging for humans, as opposed to focusing on cases which are easy for humans but challenging for AIs.

In this context, however, it might be wise to more clearly acknowledge works that also break away from the abovementioned trend. E.g. it would benefit the paper to say a few more words about the Abstraction and Reasoning Corpus (https://arxiv.org/abs/1911.01547) which was cited, but, in my opinion, without fully clarifying why it was relevant.

- The paper is very clearly written and is a pleasure to read.

- I believe that the paper has the potential to be highly impactful and of broad interest and use to large portions of AI and ML community.

**Weaknesses:**

Overall, the paper has a sound design and already acknowledges some of its limitations none of which I find to be a deal-breaker. That being said,

- The biggest limitation not fully discussed is the dataset's modest size, which, unfortunately, restricts it solely to the role of a performance benchmark rather than a potential source of training data. This limits the paper's potential impact.

- Another issue, discussed, but not fully acknowledged as a limitation, is the lack of "partial success" indicators. This is mitigated by the presence of questions with different difficulty levels, but might still be problematic. When solving a question requires a complex sequence of actions, it is highly desirable to have some measure of where the process breaks down. While the authors speculate that, potentially, in the future "the paradigm of LLMs calling tools for every task other than text understanding might not stay", the reality of today is such that having more nuanced/partial feedback would be helpful.

None of the issues listed above is a disqualifying weakness, however. I offer some suggestions on how, in my view, these issues could be mitigated in the next section.

**Questions:**

1) The authors insist that a component of human labor is crucial to create high-quality questions ("we believe human annotators are currently essential to have diverse and grounded questions, as opposed to programmatically generated ones"). However, in my opinion, this might be underselling one key idea behind their own work.

It seems that, fundamentally, the questions in GAIA are challenging because they require reverse-engineering a fairly random process of question generation, which includes randomly choosing which tool to use, what to google, and so on. So the problems are challenging for AIs because they are under-defined inverse problems.

At the same time, problem generation is a (straight)forward process. The instructions given in appendix C could, quite possibly, be adapted to tool-assisted LLM prompts. Such LLM-driven automation might offer a good compromise of data price and quality, and such approaches seem to be getting traction (e.g. https://arxiv.org/abs/2303.15056, https://arxiv.org/abs/2305.13877).

I would like to highly encourage the authors not to dismiss this direction, as adding such automation might dramatically increase the impact of their work. In the very least, it might be worth acknowledging this possibility in the discussion.

2) As a potential way to mitigate the absence of "partial success indicators", it might be useful to release a version of the dataset where one possible ground truth solution trace is explored and, potentially, in which associated files are pre-processed (e.g. images are replaced by their detailed verbal descriptions). This simplified dataset version would allow to better diagnose existing models, and to disentangle a) conceptual difficulties of aggregating information from different sources and b) procedural challenges (choosing what tool to use, what to google, and, in general, what information to collect).

---

> ### Author Response · Authors · 2023-11-20
>
> We thank the reviewer for their insightful comments.
>
>
> ## Size of the dataset
>
> We discuss this concern, that is shared with other reviewers, in our general response.
>
>
> ## Further considering automation
>
> This is a great point that we will discuss in the revised manuscript.
> - Current best models are not good enough for diverse generations given many strict constraints such as in our protocol (the most important and difficult being unambiguity in the question), which we believe is a different setup from https://arxiv.org/abs/2303.15056 and https://arxiv.org/abs/2305.13877. This is why human labor was instrumental to create valid questions. In fact, even human annotators had difficulties enforcing all the constraints in our protocol. Hence, we are pessimistic in the short term on automating generation of both varied and unambiguous questions.
> - However, the problem becomes much more tractable by removing one of the constraints. Removing the diversity constraint would allow to automatically generate unambiguous questions for a number of sources of truth limited by the human workforce needed to write associated code for each source. For example, given a source of truth such as wikipedia, one could come with an implementable set of rules leading to unambiguous questions and automatically fetch the answer. Removing the unambiguity constraint seems more scalable. In that case however, the answer, that would not be unique anymore, should probably be evaluated differently e.g. via a reward model (for the reasoning path), a judge model, or a human judge, with associated downsides in each setup.
>
> To conclude, removing the ambiguity constraint is a promising path towards automated generation of GAIA-type data for training, at the expense of making factual evaluation of the final answer challenging again.
>
>
> ## Adding indicators of partial success
>
> This is also a great point. We already release the annotator's trace for the validation set, which can be used for this purpose since it contains which tools were used, how, and very often, intermediate information required to compose the final answer. Moving forward, it would be easy to modify the protocol to systematically include such intermediate information in a way that is easy to process. However, we believe it is not desirable to release such data for the test set, as this could lead to overfitting. We will add this discussion in the manuscript.

---

> > ### Comment · Reviewer_CDvr · 2023-11-20
> > **Thank you for your response**
> >
> > Thank you for your answer. I have read other reviews and the authors' responses.
> >
> > I understand the sources of difficulties when it comes to automating the process, I just thought that it's important to discuss this possibility in the paper. I am glad that the authors are ready to do so.
> >
> > Regarding dataset size - I think it's okay to admit that "it would be great if GAIA was 10 or 100 times larger". This is a limitation/area of improvement (in the sense of "something that could have been better"), but it's not a disqualifying limitation.
> >
> >  As authors show, even at its present size, GAIA dataset is diagnostic of general model capacity. In other words, the dataset can serve its purpose "as is" and will likely be useful for the community.
> >
> > Overall, I still think it's a strong paper, and I still can confidently recommend its acceptance. Therefore I'm keeping my "8: accept, good paper" overall rating unchanged.

---

### Official Review · Reviewer_nJWa · 2023-11-06

**Soundness:** 2 fair
**Presentation:** 3 good
**Contribution:** 1 poor
**Rating:** 3
**Confidence:** 4

**Summary:**

This paper presents a new benchmark for general AI assistants, GAIA. The benchmark consists of 466 questions across 3 levels of complexity, defined by the number of steps required to solve the task. The questions are designed with three principles in mind: (1) conceptually simple, but tedious for humans. (2) Interpretability, (3) Robustness against memorization. The process for generating the dataset started with authors designing initial questions, and sharing them with annotators alongside instructions to create more questions. Then, questions were validated by ensuring that additional annotators came up with the same response  to the questions. These validated questions formed the final set, which were then fed to GPT4 with and without plugins, and AutoGPT with the ChatGPT backend. Reported results show that GAIA is challenging for these agents, while being very easy for humans. Performance is also reported on a per-capability bases, with questions divided into 5 categories: Web Browsing, Coding, Multi-modality, Diverse File-type reading, and N/A.

**Strengths:**

1. Important problem: As LLMs advance and become part of everyday life, General AI assistant benchmarks are exceedingly important.

2. Markedly different from other benchmarks: existing benchmarks focus on Narrow AI: expertise in a specific domain. GAIA focusses on more general purpose tasks that require multiple steps and are tedious rather than being hard for humans. Such simple but tedious tasks are optimally suited for AI assistants, making it a great guiding principle for the benchmark.

3. Paper easy to follow: The paper does a great job of presenting the approach, design decisions, and related work. The figures explain the work well too.

**Weaknesses:**

1. The venue is not a great fit. The majority of the contribution here is annotated dat, and the design decisions made in doing so. This would make the work a better fit for conferences focussing on these aspects, including CHI and UIST. There are no learned representations, or models, putting it out of the domain of the ICLR community.

2. Size and composition of the dataset: While GAIA is a good start, 466 questions seems like a very small dataset for a general purpose AI agent. Furthermore, most of these questions come from web browsing, which makes the benchmark quite close to a narrow AI benchmark for web browsing.

3. Experiments very thin: The utility of designing a benchmark could be justified if it taught reasonable insights about the behaviour of existing models. In its current form, the investigations presented in the paper do not offer any new insights about the behavior of these models.

**Questions:**

I don't have any particular questions, as I see a serious misfit with the conference.

---

> ### Author Response · Authors · 2023-11-20
>
> We thank the reviewer for their insightful comments.
>
>
>
> ## Fit of the venue
>
> The ICLR 2024 website lists benchmarks as a topic relevant to the conference. For example, the influential MMLU benchmark was published at ICLR 2021. Therefore, we cannot agree with the reviewer on the lack of fit for the venue. We would also like to emphasize that progress towards “learning better representation” would not be possible without clear ways to measure said progress, such as our proposed GAIA.
>
>
> ## Size and composition of the dataset
>
> We discuss this concern, that is shared with other reviewers, in our general response.
>
>
> ## Experiments
>
> To the best of our knowledge, our experiments do provide new insights into current models:  in spite of being the best available assistants equipped with relevant tools such as web browsing or image understanding (in the form of plugins), the models we evaluate do badly at our realistic yet simple tasks. In particular, Figure 4 displays the performance of LLMs on GAIA per capability. This suggests that, beyond giving LLMs access to tools, a lot more work is needed for LLMs to become viable assistants, which goes against a popular discourse in the AI community.
>
> We discuss the possibility for more detailed analysis in the general comment. The lack of more detailed insights is due to the scarcity in (i) capable models to evaluate and (ii) detailed logs of OpenAI model’s answers, including detailed tool calls. We are confident that going forward, such analysis will be possible thanks to submissions including detailed log and / or more open models. Does the reviewer have a particular experiment in mind?

---

> > ### Comment · Reviewer_CDvr · 2023-11-23
> > **Second opinion**
> >
> > I resonate with the authors' response on this matter. Benchmark/dataset papers are crucial for the ICLR community and for the advancement of the AI/ML fields in general. In this particular case, I am willing to actively advocate for the paper to be accepted and would like to encourage Reviewer nJWa to join in on the discussion (or, if their concerns were resolved, to adjust the score). It would be highly unfortunate if this high-quality contribution is rejected because of a potential misunderstanding.

---

> > ### Comment · Reviewer_nJWa · 2023-11-30
> > **Response**
> >
> > **Fit of Venue:**
> >
> > After taking into the author's response and more research into past ICLR papers, I agree that the fit of venue is not a problem.
> >
> > **Size and Composition (Responding to general comment above):**
> >
> > I disagree that the purpose is to simply rank models. IF the claim is to rank models' capabilities to be a General AI assistant, then the breadth and depth of concepts comparing them is paramount. As an example, we could create a benchmark simply consisting of music trivia questions, which could rank models. But this would not be sufficient to gauge their capabilities as a General AI Assistant. Breadth of concepts, and number of questions is paramount to be able to make such a claim.
> >
> > The bigger question is not if the questions are 466, or how many steps these questions require. The idea is to be able to make claims about how good a General AI Assistant is, we first must specify the tasks we expect a General AI assistant to complete, and then compile a benchmark for it well-representing these questions. My initial comments about venue were important because this process of understanding the goals of a General AI assistant might involve user studies, as is common in Human Computer Interaction literature. Currently, it is unclear why these 466 questions (or 5000 sub-questions) are good representatives of what we seek from a General AI Assistant.
> >
> > **Experiments:**
> >
> > The insights provided here are---(1) LLMs + Plugins are not as good as Humans. (2) Level 3 multi-step questions are harder than level-1 questions. The performance for GPT+Plugins is between 12-17% for all task subtypes (Coding, multi-modality, etc.).
> >
> > Let us focus deeper into this: On these 466 questions, GPT+Plugin achieves achieves 14% on questions involving Coding. What insight does this provide to the field? Can we say what kind of coding questions it struggles with? Do we understand better about the behavior of LLMs in terms of their coding capability? Also, for questions with multi-modality this model achieves 13%. Can we say LLMs do better with Coding then, based on the numbers on plot? Will these insights hold true for other similar question sets that may include knowledge of coding? Above all, there are no error bars, or statistical analyses provided. There are no hypotheses presented, and no hypothesis testing to support or refute these claims. So, it is unclear to me what the claims and insights are, and how the plots and analyses scientifically back these claims.
> >
> > In summary, the contribution I see here is of a set of questions that are currently beyond the capabilities of LLMs. However, I am unsure if these 466 questions reflect the tasks a General AI Assistant must complete, as these tasks are never operationalized or discussed. The only insight I get from the plots are that all models perform poorly than humans on complex questions, and plugins improve accuracy. Performance is largely poor across all sub-types, and there are no specific trends of use among these categories of questions. There are no scientific claims made or backed with data or scientific analyses presented.

---

### Author Response · Authors · 2023-11-20
**General response**

We thank the reviewers for their insightful comments and many great questions. We have responded to each reviewer’s comments separately, and are sharing a summary response covering the common threads in the reviewers’ responses.


## Size of the dataset

Reviewers commented about the relatively small size of our evaluation dataset. We argue that this is not limiting its applicability as detailed below – we will discuss those arguments in the revised version of the main paper:

- What primarily matters for a benchmark  is to effectively evaluate and rank models. Its number of questions is only one factor to serve this purpose but not an end. Therefore, the question should be “Would GAIA reflect the models performance and their abilities?”. We argue this is the case, as indicated by the scores of our evaluated models (GPT-4+Oracle >> AutoGPT+GPT-4 > GPT-4 > GPT-3.5 – note that we will add GPT-4 turbo and GPT-4 vision which are found significantly better than GPT-4, consistent with OpenAI claims).

- GAIA is meant to evaluate general models and each question can be seen as a multitask problem, combining different capabilities through multi steps (various tool use, planning, natural language understanding and generation). For level 3, questions can take up to more than 40 steps (see Figure 7). This is in contrast to more focused benchmarks, e.g. SQuAD. Therefore, when considering GAIA at its number of total sub-questions, it actually has a much larger size (~5000 in total).

- For knowledge based datasets, such as MMLU, it is critical to have a high number of questions, in order to ensure having enough questions that are not simply answered by pure memorization among all of them. Conversely we constructed GAIA questions so that they cannot be answered simply by memorization (see page 5, top “Our third principle is robustness against memorization…”).

Also, note that keeping the number of questions small makes evaluation much faster and cheaper.

Finally, the paper contribution is also about the methodology to build such evaluations, enabling the community to extend GAIA in future work.


## Detailed error analysis

We also would be quite interested in a detailed error analysis! Until now, it is unfortunately difficult to provide so mainly because it requires more detailed logs of the answers than what OpenAI API currently provides. For example, the API does not reveal which tool calls were made exactly, and what was their output. However, we have great hopes to see more analysis possible via GAIA in the near future:

- To incentivise such analysis, our leaderboard includes the possibility to report the trace of the models when generating answers, so that anyone will be able to conduct error analysis on the reported models.

- We expect that, as more models get submitted to the leaderboard, we'll be able to extract and aggregate high level information on the common failure points of models. In particular, with more and better open source models, it will become easier to track this aspect. Any in depth analysis on our current model baselines would not say anything general about tool augmented assistant LLMs, but instead be a commentary on the specific capabilities of OpenAI's closed source models at the time of writing.

---

### Meta-Review · Area_Chair_gQjx · 2023-12-07

**Metareview:**

This is a good dataset paper. The experimental results confirm that the proposed methodology leads to significant challenges for state-of-the-art LLMs.

However, I do agree with the reviewers' assessment of certain weaknesses in the paper:
- The dataset contains only 466 questions, which may not adequately represent the full spectrum of scenarios and challenges faced by GAI systems.
- It remains unclear whether the types of questions posed in this dataset can provide a comprehensive assessment of the capabilities required for General Artificial Intelligence (GAI) Assistants.

Of these weaknesses, the second point is of greater significance and impacts the overall rating of this paper. It is important that the authors address this concern by providing justification for how their benchmark aligns with the requirements of GAI Assistants. If such justification cannot be provided, I recommend that the authors consider revising the paper's title and claims to better reflect the scope and limitations of their work.

**Justification For Why Not Higher Score:**

- The dataset contains only 466 questions, which may not adequately represent the full spectrum of scenarios and challenges faced by GAI systems.
- It remains unclear whether the types of questions posed in this dataset can provide a comprehensive assessment of the capabilities required for General Artificial Intelligence (GAI) Assistants.

**Justification For Why Not Lower Score:**

This is a good dataset paper. The experimental results confirm that the proposed methodology leads to significant challenges for state-of-the-art LLMs.

---

### Decision · Program_Chairs · 2024-01-16

Accept (poster)